# TRUNCATE WITHOUT FEAR: MODULE AGGREGATION AND REDISTRIBUTION IN FEDERATED LOW-RANK ADAPTATION

## ABSTRACT

While low-rank adaptations (LoRA) have shown promise as an efficient fine-tuning technique in federated learning (FL) to reduce communication complexity, the practical application requires careful attention to the challenges posed by the aggregation schemes on client modules. In this paper, we introduce TFLoRA, which directly optimizes over the adapter weights $W = BA^\top$, and redistributes the LoRA modules using the updated adapter weights. Our theoretical analysis shows the truncation error introduced during the redistribution step is mild and TFLoRA achieves an $O(1/\sqrt{T})$ convergence rate. Compared to the existing methods, TFLoRA supports a wide range of optimizers on the server side and maintain the advantages in low communication overhead. We show empirical evidence that TFLoRA achieves better performance than the state-of-the-art federated LoRA mechanisms on various benchmarks including image/text classification and commonsense inference. Additionally, TFLoRA is demonstrated to be more favorable as the number of clients increases and with non-i.i.d client data distributions.

## 1 INTRODUCTION

Low-rank adaptation (LoRA) (Hu et al., 2021) has become popular in recent years as a supervised fine-tuning technique for large neural networks. For a linear layer in a neural network, denote $W_0 \in \mathbb{R}^{m \times n}$ as the pretrained model weight. LoRA modifies the forward pass by additively integrating a low-rank matrix $BA^\top$ into $W_0$, where $B \in \mathbb{R}^{m \times r}$, $A \in \mathbb{R}^{n \times r}$, and the rank $r \ll \min\{m, n\}$. Throughout the paper, we name the low-rank adapters $B$ and $A$ as the LoRA modules, and their product matrix $W = BA^\top$ as adapter weights. LoRA modules typically contain up to 5% of the full parameter size. By adapting solely the LoRA modules and keeping the rest of parameters frozen, pretrained neural network models, especially LLMs can be adapted to various new tasks. Such efficiency in parameters makes LoRA a promising technique to be adopted in federated learning (FL), since FL clients are usually resource-constrained and the communication cost between server and clients are particularly important (Malaviya et al., 2023).

While the LoRA paradigm is clear in the centralized setting, it remains controversial on how to apply model averaging in the federated learning context. Suppose $C$ clients participate in the federated learning. At the end of local training phase, each client maintains its local copy of LoRA modules $B^c$ and $A^c$. The server aggregates the $C$ copies into a global LoRA module while maintaining the low-rank constraints. The state-of-the-art technical roadmap bifurcates into two separate ways. FedIT (Zhang et al., 2024) and FLASC (Kuo et al., 2024) advocate directly applying the average to LoRA modules on the server side, i.e. $B = \frac{1}{C} \sum_{c=1}^{C} B^c$. This thread of approaches automatically satisfies the low rank constraints, but leads to an inexact aggregate on the adapter weight $W$ due to

$$\left( \frac{1}{C} \sum_{c=1}^{C} B^c \right) \left( \frac{1}{C} \sum_{c=1}^{C} A^c \right)^\top \neq \frac{1}{C} \sum_{c=1}^{C} B^c A^{c\top}.$$

The discrepancy is subsequently termed as *aggregation noise* in (Wang et al., 2024b). Additionally, the approach faces challenges when the client models adopt heterogeneous ranks (Cho et al., 2024). Other approaches (Singhal et al.; Wang et al., 2024b) apply averaging over the local adapter weights $W^c = B^c A^{c\top}$. Although eliminating the aggregation noise, projecting the averaged adapter weight

$W = \frac{1}{C}\sum_{c=1}^{C} W^c$ back to a low rank matrix is challenging since the rank of $W$ is at most $rC$. The existing methods unanimously find proxies for the averaged LoRA modules and transmit the discrepancy with the noise-free adapter weights to the clients, which is in the size of $m \times n$, to achieve exact updates. However, these methods lose the advantages of communication efficiency achieved via LoRA training. In addition, the existing approaches lack support on the flexibility of server optimizers. To summarize, the structure of LoRA poses unique challenges in the module merging techniques for FL and requires careful designs.

In this work, we present Truncated FLoRA (TFLoRA) that introduces a truncation step to the adapter weights to maintain low-rank LoRA modules. We theoretically show that the impact of truncation error on the convergence rate is mild. Our main technical contributions are summarized as follows.

1) We propose TFLoRA in which the gradient update is applied directly to adapter weights. To maintain the low-rank constraint on the server model, TFLoRA adopts a redistribution step to obtain updated LoRA modules with truncation on the adapter weight in the spectral domain. We showcase the difference of TFLoRA with FedIT via a simple matrix factorization problem where FedIT fails to converge to the global minima.

2) We theoretically prove TFLoRA converges to a stationary point on the LoRA modules with rate $O(1/\sqrt{T})$ under boundedness assumptions. One of the key technical observation is that the truncation error is summable over the training iterations. Additionally, we prove the boundedness of the iterates under quadratic growth condition, which leads to the same convergence rate.

3) Empirical studies on vision and language benchmarks are conducted to validate the performance of the proposed TFLoRA. We compare with the existing baselines on federated low-rank adaptation methods and demonstrate that TFLoRA is more advantageous when the number of clients is higher and the data are distributed heterogeneously among clients.

## 2 PROPOSED APPROACH

In this section, we propose Truncated FLoRA (TFLoRA) as a novel LoRA module merging mechanism that respects the aggregation noise issues and keeps the same communication efficiency as FedIT. Denote $\mathcal{L}$ as the global empirical loss, and $\mathcal{L}^c$ as the loss on client $c$. TFLoRA consists of three major steps in each training iteration. The framework is described in Algorithm 1.

**Computing Pseudo-gradient of $W$.** Under canonical federated learning settings, clients perform multiple local updates using the local dataset. The seminal work FedOPT (Reddi et al., 2020) proposes to utilize the psuedo-graident, i.e. the negative of the average model difference as the proxy of the gradient at the server model. While FedIT (Zhang et al., 2024) regards the LoRA modules $A$ and $B$ as the optimizees, and the pseudo-gradients are computed on the LoRA modules separately, our proposed TFLoRA computes the pseudo-gradient over the adapter weight matrix $W$ at the server side by $\tilde{\nabla}\mathcal{L}(W_t^l) = W_t^l - \frac{1}{C}\sum_{c=1}^{C} W_t^{l,c} = B^l A^l - \frac{1}{C}\sum_{c=1}^{C} B_{t,K}^{l,c} A_{t,K}^{l,c}{}^\top$. The pseudo-gradient $\tilde{\nabla}\mathcal{L}(W_t^l)$ characterizes the average client update on the adapter weight matrix. An analysis on the difference between the pseudo-gradient and the real gradient can be found in (Wang et al., 2024a). Through this step, we have ruled out the effect of the cross-product matrices within clients.

**Applying Server Optimizer to $W$.** We call gradient-based optimizers to perform a single optimization step on $W$, where the gradient is $\tilde{\nabla}\mathcal{L}(W^l)$ and server learning rate $\kappa$, i.e. $W_{t+1}^l = \texttt{SERVER\_OPT}(W_t^l, \tilde{\nabla}\mathcal{L}_t, \kappa)$. If the server optimizer is gradient descent (GD) with learning rate set as 1, the update on $W$ is identical to the FedAvg algorithm (McMahan et al., 2017), i.e. the average over local adapter weights. We will use $\kappa = 1$ in the subsequent theoretical analysis. TFLoRA naturally supports any adaptive optimizers including Adam (Kingma, 2014) and AMSGrad (Reddi et al., 2019), which arguably accelerate the optimization process and often achieve better generalization performance. The acceleration effect is exceptionally valuable in federated learning since it directly leads to a reduction in the overall communication costs.

**Redistribution of LoRA modules.** In this step, we project the adapter weight back into the LoRA modules, which will later be transmitted and used as an initial point at each client in the next round. Recall that the pseudo-gradient is defined as the average of the client model updates. Since the layer weight of each client is of rank $r$, the averaged weights $\bar{W}_t^l = \frac{1}{C}\sum_{c=1}^{C} B_{t,K}^{l,c} A_{t,K}^{l,c}{}^\top$ amount to at

---

**Algorithm 1:** Truncated FLoRA (TFLoRA)

---

**Input:** Server learning rate $\kappa$, Client learning rate $\eta$, Initial parameters $\{B_0^l, A_0^l\}$, LoRA rank $r$, Server Optimizer SERVER_OPT, Client Optimizer CLIENT_OPT.

**Output:** $\{B_t^l, A_t^l\}_{t=1}^T$

**for** $t = 1, 2, \ldots, T$ **do**

    **for** $c = 1, 2, \ldots, C$ **do**

        Client download LoRA parameters $\{A_t^l, B_t^l\}$

        **for** $k = 1, 2, \ldots, K$ **do**

            Perform updates by CLIENT_OPT:

            $A_{t,k}^{l,c}, B_{t,k}^{l,c} = \text{CLIENT\_OPT}(A_{t,k-1}^{l,c}, B_{t,k-1}^{l,c}, \mathcal{L}, \eta)$;

        Upload to the server: $A_{t,K}^{l,c}, B_{t,K}^{l,c}$;

    Server: Average client updates in product matrix: $\bar{W}_t^l = \frac{1}{C} \sum_{c=1}^C B_{t,K}^{l,c} A_{t,K}^{l,c}{}^\top$;

    Compute the pseudo-gradient: $\tilde{\nabla}\mathcal{L}_t = W_t^l - \bar{W}_t^l$;

    Update parameters: $W_{t+1}^l = \text{SERVER\_OPT}(W_t^l, \tilde{\nabla}\mathcal{L}_t, \kappa)$ and apply SVD to $W_{t+1}^l$:

$$U_{t+1}^l \Sigma_{t+1}^l (V_{t+1}^l)^\top = \text{SVD}(W_{t+1}^l);$$

    Truncate to rank $r$: $\bar{U}_{t+1}^l = U_{t+1}^l[:,:r]$;    $\bar{\Sigma}_{t+1}^l[:r,:] = \Sigma_{t+1}^l$;    $\bar{V}_{t+1}^l = V_{t+1}^l[:,:r]$;

    Update LoRA parameters and send to clients: $B_{t+1}^l = \bar{U}_{t+1}^l \sqrt{\bar{\Sigma}_{t+1}^l}$;   $A_{t+1}^l = \bar{V}_{t+1}^l \sqrt{\bar{\Sigma}_{t+1}^l}$;

---

most $rC$ rank, which exceeds the predefined LoRA rank $r$. The updated adapter weight $W^l$ generally possesses higher rank than $r$, which renders a lossless low-rank factorization unapproachable. It is clear that the situation is more tricky when the the number of clients is large. Nonetheless, in this step, we apply an SVD decomposition to the updated adapter weight, and we will show the truncation step can be theoretically bounded. Denote the SVD step as $U_{t+1}^l \Sigma_{t+1}^l (V_{t+1}^l)^\top = \text{SVD}(W_{t+1}^l)$, where $U_{t+1}^l \in \mathbb{R}^{m \times \min\{m,n\}}$ and $V_{t+1}^l \in \mathbb{R}^{n \times \min\{m,n\}}$ are unitary matrices, and $\Sigma_{t+1}^l$ is a diagonal matrix of $\mathbb{R}^{\min\{m,n\} \times \min\{m,n\}}$. The diagonal elements of $\Sigma_{t+1}^l$ are arranged in a descending order.

We redistribute the LoRA modules by $B_{t+1}^l = \alpha \bar{U}_{t+1}^l \sqrt{\bar{\Sigma}_{t+1}^l}$; $A_{t+1}^l = \frac{1}{\alpha} \bar{V}_{t+1}^l \sqrt{\bar{\Sigma}_{t+1}^l}$; where $\bar{U}_{t+1}^l$ and $\bar{V}_{t+1}^l$ are the top-$r$ left and right singular vectors respectively, and $\bar{\Sigma}_{t+1}^l$ are the top-$r$ singular values. The square root operator is applied element-wisely. $\alpha$ is a hyperparameter for adjusting unbalanced norms between $B$ and $A$. $\alpha$ does not affect the performance at the current iterate but will impact the subsequent optimization trajectory. For simplicity, we set $\alpha = 1$ in the theoretical analysis, while empirically we find $\alpha \geq 1$ slightly improves the model performance.

## 2.1 SHOWCASING THE DIFFERENCE OF MERGING MECHANISMS

We provide a concrete example to intuitively show how TFLoRA differs from FedIT (Zhang et al., 2024) and leads to different global models. Both methods use direct averaging on the server side.

Consider a matrix factorization problem $\min_{b \in \mathbb{R}^m, a \in \mathbb{R}^n} \mathcal{L}(b, a) = \frac{1}{2}\|ba^\top - \Sigma\|^2$, where $\Sigma$ is a rank-2 matrix with SVD decomposition $\Sigma = \sigma_1 u_1 v_1^\top + \sigma_2 u_2 v_2^\top$. $b$ and $a$ can be regarded as the LoRA modules, while the pretrained matrix $W_0$ has been integrated into $\Sigma$. Without loss of generality, assume $\sigma_1 > \sigma_2$. The data model at the two clients are $\Sigma_1 = 2\sigma_1 u_1 v_1^\top$ and $\Sigma_2 = 2\sigma_2 u_2 v_2^\top$ respectively, and the local loss $\mathcal{L}_c$ is defined as $\mathcal{L}_c(b, a) = \frac{1}{2}\|ba^\top - \Sigma_c\|^2$. One can verify that it is a valid federated learning environment since $\nabla \mathcal{L}(b, a) = \frac{1}{2}(\nabla_a \mathcal{L}_1(b, a) + \nabla_a \mathcal{L}_2(b, a))$. We follow the one-shot SGD paradigm, where the local runs are trained to convergence in which the local sequences are only exchanged once, after the local runs have converged (Mcdonald et al., 2009; Zinkevich et al., 2010). We compare the difference between $b^{\text{TFLoRA}}, \bar{a}^{\text{TFLoRA}}$ generated by Algorithm 1 and $\bar{b}^{\text{FedIT}} = \frac{1}{2}(b_1 + b_2)$ and $\bar{a}^{\text{FedIT}} = \frac{1}{2}(a_1 + a_2)$ from FedIT by the proposition.

**Proposition 2.1.** *If the LoRA modules are initialized using Gaussian distribution with mean 0 and* $\epsilon^2$, *where* $\epsilon = \tilde{O}(\frac{\sigma_2}{\sqrt{r^3 \sigma_1}(m+n)})$. *Then under one-shot SGD paradigm, with high probability over initialization,* $\mathcal{L}(\bar{b}^{\text{FedIT}}, \bar{a}^{\text{FedIT}}) \geq \mathcal{L}(\bar{b}^{\text{TFLoRA}}, \bar{a}^{\text{TFLoRA}})$.

Clearly, FedIT leads to a suboptimal solution in the specific scenario and TFLoRA outperforms.

## 2.2 CONVERGENCE ANALYSIS

We provide a convergence analysis for Algorithm 1. We consider the case when SGD is adopted as the optimizer at both server and client side. In fact, Algorithm 1 is a non-standard optimization algorithm. The theoretical challenges stem from the usage of low-rank truncation in the optimization process and the unique client averaging scheme. First, the truncation step brings up additional error and makes the trajectory of the iterates inconsistent. Second, our client averaging scheme breaks the connections between server and client optimizers, given that the clients perform parameter-efficient fine-tuning over the LoRA modules while the server directly optimizes over the adapter weights.

To show the theoretical guarantees on TFLoRA , we first make the following assumptions. All these assumptions are mild in the sense that they are defined on the adapter weights $W$, not LoRA modules. These assumptions are also used in the seminal works (Stich, 2018) in the FL literature.

**Assumption 2.2.** (Smoothness) The loss function $\mathcal{L}$ is $L$-smooth, i.e. $\|\nabla\mathcal{L}(x)-\nabla\mathcal{L}(y)\| \leq L^g\|x-y\|$. The local loss function $\mathcal{L}^c$ is $L^c$-smooth.

**Assumption 2.3.** (Bounded Gradient) The gradient of $\mathcal{L}$ with respect to $W$ is uniformly bounded, i.e. $\|\nabla\mathcal{L}(W)\| \leq G$.

**Assumption 2.4.** (Bounded Client Deviation) The difference between client gradient and the global gradient is bounded, i.e. $\|\frac{\partial\mathcal{L}^c}{\partial W} - \frac{\partial\mathcal{L}}{\partial W}\| \leq \sigma$.

We denote $L$ as the upper bound for the global and client smoothness. Additionally, due to the nonlinearity of LoRA modules (Malinovsky et al., 2024), we make the following assumption.

**Assumption 2.5.** (Bounded Iterate) The iterates $W_t$ from Algorithm 1 is bounded, i.e. $\|W_t\|^2 \leq D$.

Assumption 2.5 directly leads to the boundedness of $B_t$ and $A_t$ respectively by our LoRA module redistribution mechanism. Since $\frac{1}{\alpha}B_t$ and $\alpha A_t$ share the singular values, we have $\|B_t B_t^\top\|^2 \leq \alpha^2 D$, and $\|A_t A_t^\top\|^2 \leq \frac{1}{\alpha^2}D$. By contrast, vanilla optimization on LoRA modules does not possess this ideal property for the lack of connection in $B_t$ and $A_t$ – for example, $A_t$ can be arbitrarily small in magnitude and $B_t$, in this case, can be unbounded while not violating Assumption 2.5. We will show the boundedness of $A_t$ and $B_t$ is advantageous for our convergence analysis.

For simpler exposition, we temporarily set the local training step $K = 1$. The update of the adapter weight $W_t$ can be represented by $\bar{W}_{t+1} \cong W_t - \eta B_t B_t^\top \frac{\partial\mathcal{L}}{\partial W} - \eta\frac{\partial\mathcal{L}}{\partial W}A_t A_t^\top$, where $\cong$ omits the less important terms. One key observation is that from the previous iteration, the adapter weight $W_t$ is already a low-rank matrix. After applying a local optimization step to the LoRA modules, the updated adapter weight remains *almost* low-rank. Define the truncation error as $\epsilon_t := \bar{W}_{t+1} - B_{t+1}A_{t+1}^\top$. The following lemma provides an upper bound of $\|\epsilon_t\|$.

**Lemma 2.6.** *(Informal Version of Lemma B.2) Denote $A_t, B_t$ as the iterate at epoch t. The truncation error $\|\epsilon_t\|$ can be upper bounded by $\|\epsilon_t\| \lesssim \eta^2(\sigma^2 + 2G\sigma)\|A_t B_t^\top\|$, where $\lesssim$ omits the less important terms.*

Notice that the truncation error is at most quadratic in $\eta$ and hence will be summable over optimization iterations. From this observation, we can derive the

**Theorem 2.7.** *Under Assumption 2.2-2.5. Let $\eta = T^{-1/2}$. The output of Algorithm 1 with a single local step, i.e. $K = 1$,*

$$\frac{1}{T}\sum_{t=0}^{T-1}\left(\|\frac{\partial\mathcal{L}}{\partial B_t}\|^2 + \|\frac{\partial\mathcal{L}}{\partial A_t}\|^2\right) \leq \frac{\mathcal{M}_1}{T^{1/2}} + \frac{\mathcal{M}_2}{T^{3/2}},$$

*where $\mathcal{M}_1, \mathcal{M}_2$ are constants specified in Appendix. B.1*

Next, we show that Assumption 2.5 can be satisfied in a well-structured loss function.

**Assumption 2.8.** (Bounded Minima) Let $\mathcal{S}$ be the set of minima of the loss function $\mathcal{L}(W)$. For any $W_* \in \mathcal{S}$, $W_*$ has a uniform upper bound, i.e. $\|W_*\| \leq \|\mathcal{S}\|$.

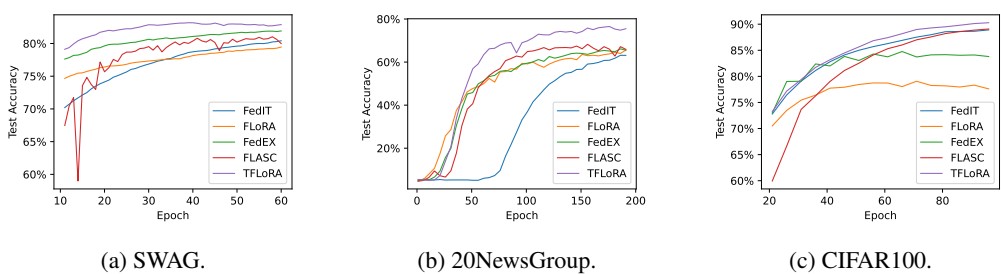

Figure 1: Test Accuracy on benchmarks. TFLoRA outperforms other federated LoRA methods.

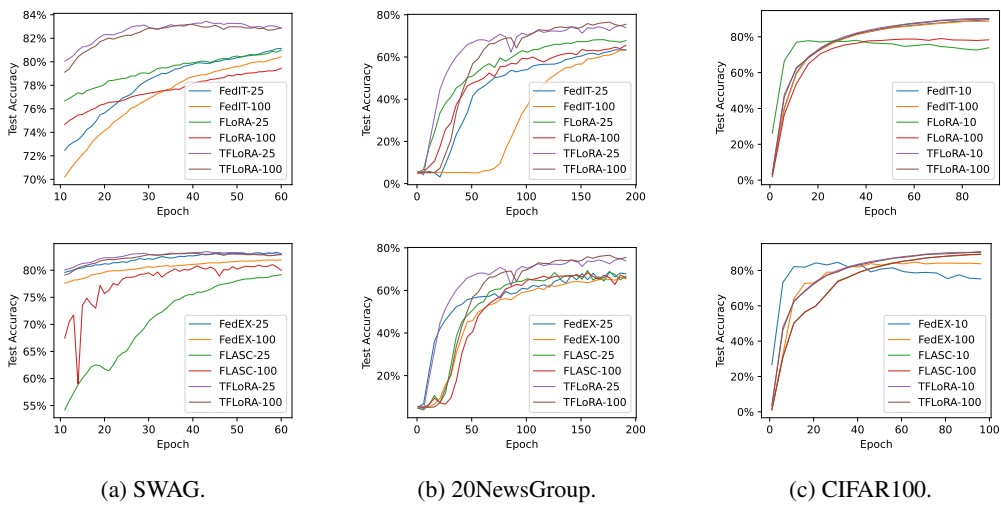

Figure 2: Test Performance with different client numbers. The digit in the legend represents the number of clients. TFLoRA degrades less in scenrios with large number of clients.

**Assumption 2.9.** (Quadratic Growth) The loss function satisfies quadratic growth condition on $W$, i.e. $\mathcal{L}(W_t) \geq \mu \text{dist}(W_t, \mathcal{S})^2$, where $\text{dist}(W_t, \mathcal{S})^2 := \min_{W \in \mathcal{S}} \|W_t - W\|^2$.

Like all previous assumptions, Assumption 2.9 is defined on the adapter weights. The assumption is weaker than a number of regularity conditions, such as, strong convexity and PL condition since Assumption 2.9 permits the existence of local minima and saddle points, which are common in non-convex optimization. In addition, Assumption 2.9 does not directly lead to Assumption 2.5 since the global loss function $\mathcal{L}$ can be unbounded As a major technical strategy, we show that along the optimization trajectory, the loss function $\mathcal{L}$ can be bounded and hence leads to boundedness on iterates. For simplicity in notation, we write the upper bound on the local gradient norm $G + \sigma$ as $\bar{G}$. We extend the convergence analysis to a multi-step local training case in the following theorem.

**Theorem 2.10.** *Suppose* $\eta = \frac{\eta_0}{2\bar{G}K\sqrt{T}}$, *subject to* $\eta_0 \leq \frac{\sqrt{\mu}\bar{G}}{18L(\|\mathcal{S}\|+M)}$. *Under Assumption 2.2-2.4, 2.8 and 2.9, the iterates of Algorithm 1 with $K$ local training steps are bounded by* $\text{dist}(W_t, \mathcal{S})^2 \leq M_2 = \frac{2}{\mu}(\mathcal{L}(W_0) + O(\eta_0))$. *The full form of constant $M_2$ can be found in Eq. 2 in Appendix C. Furthermore, the convergence rate is given by* $\frac{1}{T}\sum_{t=0}^{T-1} \|\frac{\partial \mathcal{L}}{\partial B_t}\|^2 + \|\frac{\partial \mathcal{L}}{\partial A_t}\|^2 \leq \frac{\mu \bar{G} M_2}{2\eta_0 \sqrt{T}}$.

## 3 EMPIRICAL STUDIES

In this section, we present empirical studies to validate TFLoRA on various benchmarks. We find that TFLoRA is more favorable in high client number and non-i.i.d client data distribution scenarios.

**Models and Datasets.** We incorporate three benchmarks in vision and language domains to measure the empirical performance. For the text classification problem, we choose 20newsgroup which con-

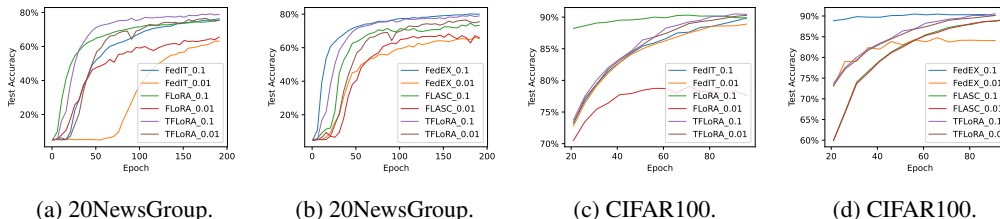

|  (a) 20NewsGroup. | (b) 20NewsGroup. | (c) CIFAR100. | (d) CIFAR100. |

Figure 3: Test performance with different levels of client data heterogeneity. The digit in the legend represents the Dirichlet prior $\alpha$. Lower $\alpha$ implies higher heterogeneity. TFLoRA is robust to the change in client data distribution.

sists of 18000 newsgroups posts on 20 topics. We finetune a GPT2 (Radford et al., 2019) with LoRA modules consisting of 2.3M parameters (1.91% of original 124M parameters). For image classification task, we adopt CIFAR-100 benchmark and finetune a ViT-Base (Vaswani, 2017) model with LoRA parameters of size 2.4M (2.84% of original 86M parameters). For the commonsense reasoning task, we utilize SWAG (Zellers et al., 2018) which consists of 113k multiple choice questions about grounded situations. We finetune a RoBERTa-Base model with LoRA parameters pf 0.6M parameters (0.47% of original 125M parameters). In all these tasks, we fix the LoRA rank $r = 16$.

**Baseline Approaches.** We compare with the existing federated LoRA variants. FedIT (Zhang et al., 2024) computes the pseudo-gradients with the averaged update on each client model and applies gradient updates to LoRA modules. FLASC (Kuo et al., 2024) builds on FedIT and transmits sparse vectors to reduce communication costs. In the experiment, we set the sparsity density of FLASC as 0.25 and set LoRA rank $r = 64$ so that the total communication bits are identical with other baseline methods. FLoRA (Wang et al., 2024b) and FedEX (Singhal et al.) transmits the discrepancy between the averaged adapter weights and the product of updated LoRA modules to ensure a *noise-less* aggregation, but at significantly higher communication costs.

**Experimental Results.** Fig. 1 shows the test accuracy achieved by the proposed and baseline methods. We set the number of clients $C = 100$ and utilize latent Dirichlet allocation (Blei et al., 2003) to distribute the dataset to clients on 20NewsGroup and CIFAR-100 datasets. We set the Dirichlet prior $\alpha$ as 0.01, which induces a high level of data heterogeneity. In all three benchmarks, TFLoRA consistently achieves the highest test accuracy. Additionally, we observe that TFLoRA converges faster than the other approaches as the accuracy curve ramps up in the few training iterations and dominates the other methods throughout the training process.

We further investigate the effect of the number of clients. We decrease the number of clients to 10 and compare the performance with other methods in Fig. 2. For smaller number of clients, TFLoRA is on par with or performs better than the baseline methods. Notably, the test accuracy of FedEX has fast initial increase in the CIFAR-100 dataset, and achieves comparable performance on the SWAG dataset. However, the test accuracy drops significantly when the client number increases to 100. In comparison, the performance degrade on TFLoRA with increased client number is much more mild.

We also investigate the effect of data heterogeneity. By increasing the Dirichlet prior $\alpha$ to 0.1, we mitigate the class imbalance among clients. In Fig. 3, we compare the performance in different levels of data heterogeneity on 20NewsGroup and CIFAR-100 datasets. It is clearly shown that almost all the methods achieve higher test accuracy when trained under milder data heterogeneity, i.e. $\alpha = 0.1$. Among all the methods, FedEX and FLoRA are the most affected by the class imbalance issues. In the 20NewsGroup dataset, FedEX achieves faster convergence and higher test performance when $\alpha = 0.1$, but falls short when $\alpha = 0.01$. In contrast, TFLoRA is robust in the sense that it can achieve a comparable or superior performance under both class imbalance conditions.

## 4 CONCLUSION

In this work, we propose TFLoRA to solve the module merging dilemma in federated low-rank adaptation. Surprisingly, we find that the truncation step, which is often regarded as a source of noise, can have mild effects in both theory and practice. We believe the findings can advance the state-of-the-art research in federated fine-tuning.

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

## A    PROOF OF PROPOSITION 2.1

We first analyze the dynamics on the clients. It directly follows from Theorem 1.1 (Ye & Du, 2021) that asymmetric low-rank matrix factorization problem converges with high probability to its global minima when the LoRA modules are initialized in the way described in Proposition 2.1, which means the local LoRA modules will not be stuck in the local minima or saddle points, for example $a = 0, b = 0$. Since we work under the one-shot paradigm, each client returns the global optima of the client loss function. The client LoRA module will be $\mathbf{b}_1 = 2\sigma_1/a_1 u_1$ and $\mathbf{a}_1 = a_1 v_1$, $\mathbf{b}_2 = 2\sigma_2/a_2 u_2$, and $\mathbf{a}_2 = a_2 v_2$. The averaged parameters are $\bar{\mathbf{b}} = \frac{\sigma_1}{a_1} u_1 + \frac{\sigma_2}{a_2} u_2$, $\bar{\mathbf{a}} = \frac{1}{2}(a_1 v_1 + a_2 v_2)$, where $a_1, a_2$ are arbitrary non-zero constants. The global risk can be written as

$$\mathcal{L}(\bar{b}^{\text{FedIT}}, \bar{a}^{\text{FedIT}}) = \frac{1}{2}\| -\sigma_1 u_1 v_1^\top - \sigma_2 u_2 v_2^\top + \sigma_1 \frac{a_2}{a_1} u_1 v_2^\top + \sigma_2 \frac{a_1}{a_2} u_2 v_1^\top \|^2.$$

On the other hand, TFLoRA first applies exact aggregation on the $\bar{W} = \sigma_1 u_1 v_1^\top + \sigma_2 u_2 v_2^\top$. Applying an SVD-truncation step will yield the leading singular pairs, i.e. $\bar{b}^{\text{TFLoRA}} \bar{a}^{\text{TFLoRA}} = \sigma_1 u_1 v_1^\top$. It is easy to show that $\mathcal{L}(\bar{b}^{\text{FedIT}}, \bar{a}^{\text{FedIT}}) \geq \mathcal{L}(\bar{b}^{\text{TFLoRA}}, \bar{a}^{\text{TFLoRA}})$.

## B    CONVERGENCE PROOF

$$B_{t+1} A_{t+1}^\top + \epsilon_t = \bar{W}_{t+1} = \frac{1}{C} \sum_{c=1}^{C} B_{t+1}^c (A_{t+1}^c)^\top$$

$$= \frac{1}{C} \sum_{c=1}^{C} (B_t - \eta \frac{\partial L^c}{\partial W} A_t)(A_t - \eta \frac{\partial L^c}{\partial W}^\top B_t)^\top$$

$$= B_t A_t^\top - \eta B_t B_t^\top \frac{\partial \mathcal{L}}{\partial W} - \eta \frac{\partial \mathcal{L}}{\partial W} A_t A_t^\top + \frac{\eta^2}{C} \sum_{c=1}^{C} \frac{\partial \mathcal{L}^c}{\partial W} A_t B_t^\top \frac{\partial \mathcal{L}^c}{\partial W}$$

$$\mathcal{L}(W_{t+1}) \leq \mathcal{L}(W_t) - \langle \frac{\partial \mathcal{L}}{\partial W}, \eta B_t B_t^\top \frac{\partial \mathcal{L}}{\partial W} + \eta \frac{\partial \mathcal{L}}{\partial W} A_t A_t^\top - \frac{\eta^2}{C} \sum_{c=1}^{C} \frac{\partial \mathcal{L}^c}{\partial W} A_t B_t^\top \frac{\partial \mathcal{L}^c}{\partial W} - \epsilon_t \rangle$$

$$+ \frac{L}{2} \| \eta B_t B_t^\top \frac{\partial \mathcal{L}}{\partial W} + \eta \frac{\partial \mathcal{L}}{\partial W} A_t A_t^\top - \frac{\eta^2}{C} \sum_{c=1}^{C} \frac{\partial \mathcal{L}^c}{\partial W} A_t B_t^\top \frac{\partial \mathcal{L}^c}{\partial W} - \epsilon_t \|^2$$

**Lemma B.1.** *Let $A$ be an arbitrary matrix and $B$ be a matrix of rank at most $r$. Let $A_r$ be the SVD approximation of $A$ with rank $r$. The truncation error $\|A - A_r\| \leq \|A - B\|$.*

We use the following lemma to quantify the truncation error term.

**Lemma B.2.** *Denote $A_t, B_t$ as the iterate at epoch $t$. The truncation error $\|\epsilon_t\|$ can be upper bounded by*

$$\|\epsilon_t\| \leq \frac{\eta^2}{C} \sum_{c=1}^{C} \|(\frac{\partial \mathcal{L}^c}{\partial W} - \frac{\partial \mathcal{L}}{\partial W}) A_t B_t^\top (\frac{\partial \mathcal{L}^c}{\partial W} - \frac{\partial \mathcal{L}}{\partial W}) + \frac{\partial \mathcal{L}}{\partial W} A_t B_t^\top (\frac{\partial \mathcal{L}^c}{\partial W} - \frac{\partial \mathcal{L}}{\partial W}) + (\frac{\partial \mathcal{L}^c}{\partial W} - \frac{\partial \mathcal{L}}{\partial W}) A_t B_t^\top \frac{\partial \mathcal{L}}{\partial W}\|.$$

*Proof.* The truncated term at epoch $t$ writes as

$$B_t A_t^\top - \eta B_t B_t^\top \frac{\partial \mathcal{L}}{\partial W} - \eta \frac{\partial \mathcal{L}}{\partial W} A_t A_t^\top + \frac{\eta^2}{C} \sum_{c=1}^{C} \frac{\partial \mathcal{L}^c}{\partial W} A_t B_t^\top \frac{\partial \mathcal{L}^c}{\partial W}$$

$$= (B_t - \eta \frac{\partial \mathcal{L}}{\partial W} A_t)(A_t^\top - \eta B_t^\top \frac{\partial \mathcal{L}}{\partial W}) - \eta^2 \frac{\partial \mathcal{L}}{\partial W} A_t B_t^\top \frac{\partial \mathcal{L}}{\partial W} + \frac{\eta^2}{C} \sum_{c=1}^{C} \frac{\partial \mathcal{L}}{\partial W} A_t B_t^\top \frac{\partial \mathcal{L}}{\partial W}$$

$$+ \frac{\eta^2}{C} \sum_{c=1}^{C} (\frac{\partial \mathcal{L}^c}{\partial W} - \frac{\partial \mathcal{L}}{\partial W}) A_t B_t^\top (\frac{\partial \mathcal{L}^c}{\partial W} - \frac{\partial \mathcal{L}}{\partial W}) + (\frac{\partial \mathcal{L}^c}{\partial W} - \frac{\partial \mathcal{L}}{\partial W}) A_t B_t^\top \frac{\partial \mathcal{L}}{\partial W} + \frac{\partial \mathcal{L}}{\partial W} A_t B_t^\top (\frac{\partial \mathcal{L}^c}{\partial W} - \frac{\partial \mathcal{L}}{\partial W})$$

By applying Lemma B.1, we get the result.    □

## B.1 PROOF OF THEOREM 2.7

*Proof.* By smoothness on $\mathcal{L}$, we can derive the $T$-step descent rule

$$\mathcal{L}(W_{T+1}) \leq \mathcal{L}(W_0) - \sum_{t=0}^{T} \eta \|B_t^\top \frac{\partial \mathcal{L}}{\partial W}\|^2 + \eta\|\frac{\partial \mathcal{L}}{\partial W} A_t\|^2 + \frac{\eta^2}{C} \sum_{c=1}^{C} \sum_{t=0}^{T} \|\frac{\partial \mathcal{L}}{\partial W}\| \|\frac{\partial \mathcal{L}^c}{\partial W}\|^2 \|A_t B_t^\top\| + \sum_{t=0}^{T} \|\frac{\partial \mathcal{L}}{\partial W}\| \|\epsilon_t\|$$

$$+ 2\eta^2 L \sum_{t=0}^{T} (\|B_t B_t^\top\|^2 + \|A_t A_t^\top\|^2) \|\frac{\partial \mathcal{L}}{\partial W}\|^2 + \frac{2\eta^4 L}{C} \sum_{c=1}^{C} \sum_{t=0}^{T} \|\frac{\partial \mathcal{L}^c}{\partial W}\|^4 \|A_t B_t^\top\|^2 + L\|\epsilon_t\|^2$$

$$\leq \mathcal{L}(W_0) - \sum_{t=0}^{T} \eta \|B_t^\top \frac{\partial \mathcal{L}}{\partial W}\|^2 + \eta\|\frac{\partial \mathcal{L}}{\partial W} A_t\|^2 + \eta^2 \sum_{t=0}^{T} G(G+\sigma)^2 \|A_t B_t^\top\| + \eta^2 \sum_{t=0}^{T} (G\sigma^2 + 2G^2\sigma) \|A_t B_t^\top\|$$

$$+ 2\eta^2 L \sum_{t=0}^{T} (\|B_t B_t^\top\|^2 + \|A_t A_t^\top\|^2) G^2 + 2\eta^4 L \sum_{t=0}^{T} (G+\sigma)^4 \|A_t B_t^\top\|^2 + 2\eta^4 L \sum_{t=0}^{T} (G\sigma^2 + 2G^2\sigma)^2 \|A_t B_t^\top\|^2$$

$$\leq \mathcal{L}(W_0) - \sum_{t=0}^{T} \eta \|B_t^\top \frac{\partial \mathcal{L}}{\partial W}\|^2 + \eta\|\frac{\partial \mathcal{L}}{\partial W} A_t\|^2 + \eta^2 \sum_{t=0}^{T} G(G+\sigma)^2 D^{\frac{1}{2}} + \eta^2 \sum_{t=0}^{T} (G\sigma^2 + 2G^2\sigma) D^{\frac{1}{2}}$$

$$+ 2\eta^2 L \sum_{t=0}^{T} 2DG^2 + 2\eta^4 L \sum_{t=0}^{T} (G+\sigma)^4 D + 2\eta^4 L \sum_{t=0}^{T} (G\sigma^2 + 2G^2\sigma)^2 D$$

where the last inequality utilizes the orthogonality of $B$ and $A$ matrix, and hence $\|B_t B_t^\top\| = \|B_t A_t^\top\| = \|A_t A_t^\top\| = \|\Sigma_t\|$.

Then rearranging the terms yields

$$\frac{1}{T} \sum_{t=0}^{T} \|\frac{\partial \mathcal{L}}{\partial B_t}\|^2 + \eta\|\frac{\partial \mathcal{L}}{\partial A_t}\|^2 \leq \frac{\mathcal{L}(W_0)}{\sqrt{T}} + \frac{G(G+\sigma)^2 D^{1/2} + (G\sigma^2 + 2G^2\sigma)D^{1/2} + 4LDG^2}{\sqrt{T}}$$

$$+ \frac{2L(G+\sigma)^4 D + 2L(G\sigma^2 + 2G^2\sigma)^2 D}{T^{3/2}}$$

Define the constants

$$\mathcal{M}_1 = cL(W_0) + G(G+\sigma)^2 D^{1/2} + (G\sigma^2 + 2G^2\sigma)D^{1/2} + 4LDG^2$$

$$\mathcal{M}_2 = 2L(G+\sigma)^4 D + 2L(G\sigma^2 + 2G^2\sigma)^2 D$$

□

**Assumption B.3.** (Quadratic Growth) The loss function satisfies quadratic growth condition on $W$, i.e. $\mathcal{L}(W_t) \geq \mu\text{dist}(W_t, \mathcal{S})^2$ where $\mathcal{S}$ is the set of optimum.

We use the following lemma to show the boundedness of the iterate under the quadratic growth condition.

**Lemma B.4.** *Let the step size* $\eta = \frac{\eta_0}{\sqrt{T_{\max}}}$. *Under Assumption B.3, the iterate generated by Algorithm 1 is bounded, i.e.* $\text{dist}(W_t, \mathcal{S})^2 \leq M$.

*Proof.*

$$\mathcal{L}(W_{T+1}) \leq \mathcal{L}(W_0) - \sum_{t=0}^{T} \eta \|B_t^\top \frac{\partial \mathcal{L}}{\partial W}\|^2 + \eta\|\frac{\partial \mathcal{L}}{\partial W} A_t\|^2 + \frac{\eta^2}{C} \sum_{c=1}^{C} \sum_{t=0}^{T} \|\frac{\partial \mathcal{L}}{\partial W}\| \|\frac{\partial \mathcal{L}^c}{\partial W}\|^2 \|A_t B_t^\top\| + \sum_{t=0}^{T} \|\frac{\partial \mathcal{L}}{\partial W}\| \|\epsilon_t\|$$

$$+ 2\eta^2 L \sum_{t=0}^{T} (\|B_t B_t^\top\|^2 + \|A_t A_t^\top\|^2) \|\frac{\partial \mathcal{L}}{\partial W}\|^2 + \frac{2\eta^4 L}{C} \sum_{c=1}^{C} \sum_{t=0}^{T} \|\frac{\partial \mathcal{L}^c}{\partial W}\|^4 \|A_t B_t^\top\|^2 + L\|\epsilon_t\|^2$$

$$\leq \mathcal{L}(W_0) + \eta^2 \sum_{t=0}^{T} G(G+\sigma)^2 \|A_t B_t^\top\| + \eta^2 \sum_{t=0}^{T} (G\sigma^2 + 2G^2\sigma) \|A_t B_t^\top\|$$

$$+ 2\eta^2 L \sum_{t=0}^{T} (\|B_t B_t^\top\|^2 + \|A_t A_t^\top\|^2) G^2 + 2\eta^4 L \sum_{t=0}^{T} (G+\sigma)^4 \|A_t B_t^\top\|^2 + 2\eta^4 L \sum_{t=0}^{T} (G\sigma^2 + 2G^2\sigma)^2 \|A_t B_t^\top\|^2$$

$$\|A_t B_t\| = \|W_t\| \le \|W_t - W^*\| + \|W^*\| \le \text{dist}(W_t, \mathcal{S})^2 + \|\mathcal{S}\| + 1$$
$$\|B_t B_t^\top\|^2 = \|A_t A_t^\top\|^2 = \|A_t B_t\|^2 = \|W_t\|^2 \le 2\text{dist}(W_t, \mathcal{S})^2 + 2\|\mathcal{S}\|^2$$

If $\text{dist}(W_t, \mathcal{S})^2 < M$ for $t \le T$, we want

$$\text{dist}(W_{T+1}, \mathcal{S})^2 \le \frac{\mathcal{L}(W_{T+1})}{\mu} \le M,$$

which translates to

$$\mathcal{L}(W_0) + \eta_0^2 (G(G+\sigma)^2 + G\sigma^2 + 2G^2\sigma)(M + \|W^*\| + 1) + 2\eta_0^2 L G^2 (M + \|W^*\|^2)$$
$$+ \eta_0^4 L((G+\sigma)^4 + (G\sigma^2 + 2G^2\sigma)^2) M \le \mu M.$$

The above condition is satisfied when

$$\eta_0^2 G(G+\sigma)^2 + 2\eta_0^2 L G^2 + \eta_0^4 L((G+\sigma)^4 + (G\sigma^2 + 2G^2\sigma)^2) \le \frac{\mu}{2},$$
$$\frac{\mu}{2} M \ge \mathcal{L}(W_0) + \eta_0^2 (G(G+\sigma)^2 + G\sigma^2 + 2G^2\sigma)(\|W^*\| + 1) + 2\eta_0^2 L G^2 (\|W^*\|^2) \quad (1)$$

$\square$

Under the boundedness condition of $A$ and $B$, we can prove the convergence of Algorithm 1.

$$\eta \sum_{t=0}^{T} \|\frac{\partial \mathcal{L}}{\partial A}\|^2 + \|\frac{\partial \mathcal{L}}{\partial B}\|^2$$

$$\le \mathcal{L}(W_0) - \mathcal{L}(W_{T+1}) + \eta^2 \sum_{t=0}^{T} G(G+\sigma)^2 \|A_t B_t^\top\| + \eta^2 \sum_{t=0}^{T} (G\sigma^2 + 2G^2\sigma) \|A_t B_t^\top\|$$

$$+ \eta^2 L \sum_{t=0}^{T} (\|B_t B_t^\top\|^2 + \|A_t A_t^\top\|^2) G^2 + \eta^4 L \sum_{t=0}^{T} (G+\sigma)^4 \|A_t B_t^\top\|^2 + \eta^4 L \sum_{t=0}^{T} (G\sigma^2 + 2G^2\sigma)^2 \|A_t B_t^\top\|^2$$

$$\frac{1}{T} \sum_{t=0}^{T} \|\frac{\partial \mathcal{L}}{\partial A}\|^2 + \|\frac{\partial \mathcal{L}}{\partial B}\|^2$$

$$\le \frac{\mathcal{L}(W_0) - \mathcal{L}(W_{T+1})}{\eta T} + \frac{\eta}{T} \sum_{t=0}^{T} G(G+\sigma)^2 \|A_t B_t^\top\| + \frac{\eta}{T} \sum_{t=0}^{T} (G\sigma^2 + 2G^2\sigma) \|A_t B_t^\top\|$$

$$+ \frac{\eta L}{T} \sum_{t=0}^{T} (\|B_t B_t^\top\|^2 + \|A_t A_t^\top\|^2) G^2 + \frac{\eta^3 L}{T} \sum_{t=0}^{T} (G+\sigma)^4 \|A_t B_t^\top\|^2 + \frac{\eta^3 L}{T} \sum_{t=0}^{T} (G\sigma^2 + 2G^2\sigma)^2 \|A_t B_t^\top\|^2$$

$$\le \frac{\mathcal{L}(W_0) - \mathcal{L}(W_{T+1})}{\eta_0 \sqrt{T}} + \frac{\eta_0}{\sqrt{T}} (G(G+\sigma)^2 + G\sigma^2 + 2G^2\sigma)(M + \|W^*\| + 1)$$

$$+ \frac{2\eta_0 L G^2}{\sqrt{T}} (M + \|W^*\|^2) + \frac{\eta_0^3 L}{T^{3/2}} ((G+\sigma)^4 + (G\sigma^2 + 2G^2\sigma)^2) M$$

## C  CONVERGENCE FOR MULTIPLE LOCAL STEPS

The chain rule on Hessian

$$H_{\mathcal{L}}(A) = (\frac{\partial W}{\partial A}) H_{\mathcal{L}}(W) (\frac{\partial W}{\partial A})^\top + \sum_i \frac{\partial \mathcal{L}(W)}{\partial W_i} H_{W_i}(B)$$

The update on the local LoRA modules are

$$B_{t+1}^c = B_t - \eta \sum_{k=1}^K \frac{\partial \mathcal{L}^c}{\partial B_{t,k}^c} = B_t - \eta K \frac{\partial \mathcal{L}^c}{\partial B_t} - \eta \sum_{k=1}^K (\frac{\partial \mathcal{L}^c}{\partial B_{t,k}^c} - \frac{\partial \mathcal{L}^c}{\partial B_t})$$

$$A_{t+1}^c = A_t - \eta \sum_{k=1}^K \frac{\partial \mathcal{L}^c}{\partial A_{t,k}^c} = A_t - \eta K \frac{\partial \mathcal{L}^c}{\partial A_t} - \eta \sum_{k=1}^K (\frac{\partial \mathcal{L}^c}{\partial A_{t,k}^c} - \frac{\partial \mathcal{L}^c}{\partial A_t})$$

The norm of LoRA modules

$$\|B_{t+1}^c\| \leq \|B_t\| + \eta G \sum_{k=1}^K \|A_{t,k}^c\| \leq D_t$$

is satisfied when we set $D_t = 2\|A_t\|$ and $\eta \leq \frac{1}{2GK}$.

Then

$$B_{t+1} A_{t+1}^\top + \epsilon_t = \bar{W}_{t+1} = \frac{1}{C} \sum_{c=1}^C B_{t+1}^c (A_{t+1}^c)^\top$$

$$= \frac{1}{C} \sum_{c=1}^C (B_t - \eta K \frac{\partial \mathcal{L}^c}{\partial B_t} - \eta \sum_{k=1}^K (\frac{\partial \mathcal{L}^c}{\partial B_{t,k}^c} - \frac{\partial \mathcal{L}^c}{\partial B_t}))(A_t - \eta K \frac{\partial \mathcal{L}^c}{\partial A_t} - \eta \sum_{k=1}^K (\frac{\partial \mathcal{L}^c}{\partial A_{t,k}^c} - \frac{\partial \mathcal{L}^c}{\partial A_t}))^\top$$

$$= B_t A_t^\top - \eta K \frac{\partial \mathcal{L}}{\partial B_t} A_t^\top - \eta K B_t \frac{\partial \mathcal{L}}{\partial A_t}^\top - \eta \frac{1}{C} B_t \sum_{c=1}^C \sum_{k=1}^K (\frac{\partial \mathcal{L}^c}{\partial A_{t,k}^c} - \frac{\partial \mathcal{L}^c}{\partial A_t})^\top$$

$$- \eta \frac{1}{C} \sum_{c=1}^C \sum_{k=1}^K (\frac{\partial \mathcal{L}^c}{\partial B_{t,k}^c} - \frac{\partial \mathcal{L}^c}{\partial B_t}) A_{t+1}^c + \eta^2 K \frac{1}{C} \sum_{c=1}^C \frac{\partial \mathcal{L}^c}{\partial B_t} \sum_{k=1}^K \frac{\partial \mathcal{L}^c}{\partial A_{t,k}^c}$$

Consider the term

$$\|\sum_{k=1}^K (\frac{\partial \mathcal{L}^c}{\partial B_{t,k}^c} - \frac{\partial \mathcal{L}^c}{\partial B_t}) A_{t+1}^c\|$$

$$\leq \sum_{k=1}^K \|H_{\mathcal{L}^c}(B_t)(B_{t,k}^c - B_t)\| \|A_{t+1}^c\|$$

$$\leq \eta \sum_{k=1}^K \|H_{\mathcal{L}^c}(B_t)\| \|\sum_{\tau=1}^k \frac{\partial \mathcal{L}}{\partial W} A_{t,\tau}^c\| \|A_{t+1}^c\|$$

$$\leq \eta K^2 L \|B_{t,k}^c\|^2 G \|A_{t,k}^c\| \|A_{t+1}^c\|$$

Consider the loss on $W$

$$\mathcal{L}(W_{t+1}) \leq \mathcal{L}(W_t) - \eta K \|\frac{\partial \mathcal{L}}{\partial A_t}\|^2 - \eta K \|\frac{\partial \mathcal{L}}{B_t}\|^2 + 2\eta G \|B_t\| \eta K^2 L D_t^2 G D_t$$

$$+ 2\eta^2 G K^2 L D_t^2 G D_t^2 + 2\eta^2 G K \|A_t\| G K D_t G + \eta^2 G K^2 G^2 \|B_t A_t^\top\|$$

$$+ 3L(\eta^2 K^2 G^2 \|A_t\|^4 + \eta^2 K^2 G^2 \|B_t\|^4 + 4\eta^4 \|B_t\|^2 K^4 L^2 D_t^6 G^2)$$

$$+ 3L(4\eta^4 K^4 L^2 D_t^8 G^2 + 4\eta^4 G^6 K^4 \|A_t\|^2 D_t^2 + \eta^4 K^4 G^4 \|B_t A_t^\top\|^2)$$

$$\leq \mathcal{L}(W_t) - \eta K \|\frac{\partial \mathcal{L}}{\partial A_t}\|^2 - \eta K \|\frac{\partial \mathcal{L}}{B_t}\|^2 + \frac{4\eta_0^2}{T} L \|B_t\|^4 + \frac{8\eta_0^2}{T} L \|B_t\|^4 + \frac{\eta_0^2}{T} G \|A_t\|^2 + \frac{\eta_0^2}{4T} G \|A_t\|^2$$

$$+ 3L(\frac{\eta_0^2}{2T} \|A_t\|^4 + \frac{16\eta_0^2}{G^2 T} \|B_t\|^8 L^2 + \frac{64\eta_0^2}{G^2 T} \|B_t\|^8 L^2 + \frac{\eta_0^2}{T} G^2 \|A_t\|^4 + \frac{\eta_0^2}{16T} \|A_t\|^4)$$

Set $\eta_0 \leq \frac{\sqrt{\eta_0'}}{\|S\| + M}$. The RHS

$$\text{RHS} \leq \mathcal{L}(W_t) + \frac{4\eta_0' L}{T} + \frac{5\eta_0' G}{4TM} + 3L(\frac{\eta_0'}{2T} + \frac{80\eta_0' L^2}{G^2 T} \|W_t\|^2 + \frac{\eta_0'}{T} G^2 + \frac{\eta_0'}{16T})$$

Aggregate the inequality by time step $t$, we require the following relationship holds

$$\mathcal{L}(W_1) + 4\eta'_0 L + \frac{5\eta'_0 G}{4M} + 3L(\frac{\eta'_0}{2} + \frac{160\eta'_0 L^2}{G^2}(M + \|\mathcal{S}\|) + \eta'_0 G^2 + \frac{\eta'_0}{16}) \leq \mu M$$

Let $\eta_0 \leq \frac{\sqrt{\mu}G}{18L(\|\mathcal{S}\|+M)}$, and

$$M = \frac{2}{\mu}(\mathcal{L}(W_1) + 4\eta'_0 L + \frac{5\eta'_0 G}{4} + 3L(\frac{\eta'_0}{2} + \frac{160\eta'_0 L^2}{G^2}\|\mathcal{S}\| + \eta'_0 G^2 + \frac{\eta'_0}{16})) \tag{2}$$

satisfies the condition.

## D    DISCUSSIONS ON COMPUTATIONAL OVERHEAD

While TFLoRA involves additional operations on the server side, in this section, we show that computational overhead is affordable in practice. The additional cost mainly comes from the pseudo-gradient computation step and the LoRA module redistribution step. To compute the pseudo-gradient, we apply matrix multiplication to the LoRA modules which amounts to $O(mnr)$ flops. For the redistribution step, we apply SVD decomposition to the adapter weights $W$ and only keeps the top-$r$ singular vectors. The operation can be efficiently implemented via Lanczos method (Lehoucq et al., 1998), which takes $O(mnr)$ flops. Notice that these operations are executed layer-wise, and hence the matrix shapes $m$ and $n$ occurring in the computational complexity are typically in thousands. For example, GPT2 (Radford et al., 2019) has $m = 3072$, $n = 768$ and for RoBERTa-Base (Liu et al., 2019) and ViT-Base (Vaswani, 2017), we have $m = n = 768$. In addition, since there is no temporal dependence on the operations and the LoRA modules of different layers typically have the same size, the matrix multiplication and SVD operations can be computed in batches and in a parallel way. Finally, all these operations requiring higher computational overhead happen on the server side, which is commonly reckoned to have abundant computing resources.

