# OpenReview forum: "Truncate without Fear: Module Aggregation and Redistribution in Federated Low-Rank Adaptation"
_ICLR.cc/2025/Workshop/MCDC — MCDC @ ICLR 2025_

### Official Review · Reviewer_RY9w · 2025-02-19

**Rating:** 6
**Confidence:** 3
**Fit:** 4

**Summary:**

Thanks for the interesting paper. This paper proposes an aggregation mechanism for federated low-rank adaptation. Utilizing psudogradient updates from FedOPT (Reddi et al., 2021), the central theme is to perform a truncation on the server-side via SVD. By Eckart-Young, it is clear that this is the best low-rank approximation of the averaged low-rank updates with respect to Frobenius norm, which are in practice higher rank. The authors then give a convergence analysis of this proposed framework. They empirically validate their approach via GPT2, ViT-B, RoBERTa-Base federated low-rank fine-tuning, which appear to show improvements over other baselines.

**Reason For Giving A Higher Score:**

The paper is a clear read, and well-motivated.

**Reason For Giving A Lower Score:**

NA

**Strengths And Weaknesses:**

The convergence analysis seems reasonably solid. The experiments use larger models appropriate and pertinent to the proposed work. The discussion is well-cited, and it is easy to see where the assumptions come from, as well as the motivation.

I am not too familiar with the federated LoRA literature. However, their paper was an interesting read from a layperson's perspective. I did not carefully check all the details of the mathematics.

**Suggestions:**

My personal preference is to have additional evaluation, especially for language models. For example, it would be interesting to see federated language model inference performance via low-rank adaptation, such as by evaluating superglue with Roberta. However, this is a strictly personal preference.

Also, I believe that the recent paper: Federated LLMs Fine-tuned with Adaptive Importance-Aware LoRA (Yang et al) may be relevant to this work.

---

### Official Review · Reviewer_1AG8 · 2025-02-27

**Rating:** 7
**Confidence:** 3
**Fit:** 4

**Summary:**

The paper introduces TFLoRA, a federated learning method that directly optimizes the adapter weight matrix \(W = BA^\top\) to avoid the aggregation noise incurred when averaging the low-rank matrices \(B\) and \(A\) separately. Instead of forming \(\bar{B}\,\bar{A}^\top\) from client updates, TFLoRA aggregates the individual adapter weights and then employs truncated singular value decomposition (SVD) to project the result back to a low-rank form. The authors provide theoretical guarantees showing that the truncation error remains mild - being at most quadratic in the learning rate - and that the method converges at an \(O(1/\sqrt{T})\) rate under standard assumptions. Empirical studies on vision and language benchmarks further demonstrate that TFLoRA outperforms state-of-the-art federated LoRA approaches, particularly in settings with high client numbers and non-i.i.d. data distributions.

**Reason For Giving A Higher Score:**

The work introduces a novel and well-motivated approach to address the challenges of federated low-rank adaptation. The comprehensive theoretical analysis, convergence proofs, and extensive empirical evaluations across diverse benchmarks collectively strengthen the paper’s contributions.

The method demonstrates clear improvements in test accuracy and robustness, particularly in highly heterogeneous data settings, which is highly valuable for real-world federated learning applications.

**Reason For Giving A Lower Score:**

Despite its strengths, the presentation in some sections, especially the dense theoretical derivations, could be more accessible. This may limit the paper’s impact on a broader audience not deeply familiar with federated learning or low-rank adaptation techniques.

The discussion on computational overhead, while addressed, would benefit from more detailed analysis on scalability in practical, large-scale deployments.

**Strengths And Weaknesses:**

Strengths:
- Novel aggregation technique -  TFLoRA most certainly is a novel technique

- Rigorous theoretical analysis - the work provides solid theoretical guarantees, including a convergence rate of
  \[
  O\left(\frac{1}{\sqrt{T}}\right),
  \]

- Comprehensive empirical evaluation - extensive experiments across multiple benchmarks (image/text classification and commonsense inference) give a good initial demonstration that TFLoRA outperforms existing federated LoRA methods. The method shows robustness against increasing client numbers and data heterogeneity.

- Flexibility with server optimizers - TFLoRA supports a variety of server-side optimizers, including adaptive methods like Adam. This flexibility is advantageous in federated learning settings, where the choice of optimizer can significantly impact communication efficiency and convergence behavior.

Weaknesses:
- Increased computational overhead - although this submission argues SVD-based truncation step is computationally efficient (e.g. using methods like Lanczos), it will likely introduce additional overhead on the server side. This might become a bottleneck in very large-scale or resource-constrained deployments.

- Has a strong reliance on theoretical assumptions - the convergence analysis is built on several assumptions (e.g., smoothness, bounded gradients, quadratic growth) that may not always be satisfied in practice, especially in highly non-convex deep learning scenarios

**Suggestions:**

The paper could benefit from clearer exposition in the theoretical sections. A brief, intuitive overview or diagram illustrating the core idea behind TFLoRA and its truncation mechanism would help readers grasp the approach without getting lost in technical details.

Additionally, including an ablation study on key hyperparameters—especially the role of the redistribution hyperparameter α—could further substantiate the practical advantages of the method.

---

### Official Review · Reviewer_cd6c · 2025-02-27

**Rating:** 7
**Confidence:** 5
**Fit:** 4

**Summary:**

TFLoRA is a method that improves LoRA in federated finetuning. While existing baselines directly average the low-rank matrices A and B, this results in aggregation noise i.e. does not match the result of first expanding the low rank updates and aggregating in the full-rank space. Other baselines attempt to compute the full rank 'noiseless' update, but this can make future communication expensive. To balance these two limitations, TFLoRA computes the full-rank aggregate update, but then uses truncated SVD to project this update back into the low-rank space. Across 3 datasets, TFLoRA achieves better performance than several other LoRA + FL baselines.

**Reason For Giving A Higher Score:**

The authors present a reasonable problem (noisy aggregation in LoRA), constraints (full rank aggregation costs more communication), and a suitable solution (projecting full rank aggregates back into the low-rank space).

**Reason For Giving A Lower Score:**

n/a

**Strengths And Weaknesses:**

The method is simple and performs well. The experiments test several standard methods and tasks.

**Suggestions:**

Could TFLoRA also improve over baselines in settings where the rank is heterogeneous? For example, Heterogeneous LoRA (Cho et al) proposes a similar low-rank aggregation scheme also suffers from aggregation noise.

The paper makes an interesting point that although TFLoRA introduces truncation noise, this error can be accumulated across multiple rounds at the server. Is there a way we can empirically compare truncation noise versus low-rank aggregation noise, and that less error w.r.t to a full-rank aggregation correlates with better performance? Can we also show that accumulating the truncation error across rounds (which appears built into the method) improves over a naive baseline that does not consider error accumulation?

Can authors provide theoretical justification, ablations, or intuition on why TFLoRA would scale much better with number of clients / heterogeneity than existing baselines?

---

### Decision · Program_Chairs · 2025-03-06

**Decision:**

Accept

**Comment:**

This work proposes a distributed adapter method, which is doubly relevant to this workshop through its modular nature of adapters and distributed optimization. The reviewers recommend acceptance and we're happy to accept it to this workshop.